# Multi-Objective Evolutionary Design of an Electric Vehicle Chassis

**DOI:** 10.3390/s20133633

**Published:** 2020-06-28

**Authors:** Pablo Luque, Daniel A. Mántaras, Álvaro Maradona, Jorge Roces, Luciano Sánchez, Luis Castejón, Hugo Malón

**Affiliations:** 1Department of Transportation Engineering, University of Oviedo, 33203 Gijón, Spain; luque@uniovi.es (P.L.); mantaras@uniovi.es (D.A.M.); UO237527@uniovi.es (Á.M.); rocesjorge@uniovi.es (J.R.); 2Department of Computer Science, University of Oviedo, 33203 Gijón, Spain; 3Department of Mechanical Engineering, University of Zaragoza, 50018 Zaragoza, Spain; luiscast@unizar.es (L.C.); hml@unizar.es (H.M.)

**Keywords:** chassis optimization, electric vehicle (EV), energy consumption, genetic algorithm, electric powertrain

## Abstract

An iterative algorithm is proposed for determining the optimal chassis design of an electric vehicle, given a path and a reference time. The proposed algorithm balances the capacity of the battery pack and the dynamic properties of the chassis, seeking to optimize the tradeoff between the mass of the vehicle, its energy consumption, and the travel time. The design variables of the chassis include geometrical and inertial values, as well as the characteristics of the powertrain. The optimization is constrained by the slopes, curves, grip, and posted speeds of the different sections of the track. Particular service constraints are also considered, such as limiting accelerations due to passenger comfort or cargo safety. This methodology is applicable to any vehicle whose route and travel time are known in advance, such as delivery vehicles, buses, and race cars, and has been validated using telemetry data from an internal combustion rear-wheel drive race car designed for hill climb competitions. The implementation of the proposed methodology allows to reduce the weight of the battery pack by up to 20%, compared to traditional design methods.

## 1. Introduction

Every year, the European Union, USA, Japan, among others, set stricter targets for CO_2_ emissions for passenger cars and commercial vehicles. This fact, combined with the rising price of petrol per barrel, social awareness regarding global warming, and many other factors, has led to a growing demand for alternatives to traditional internal combustion engine vehicles [1], such as electric, hybrid, and fuel cell vehicles.

There are numerous examples in which an electric vehicle can be used in a very efficient way; for example, in delivery trucks, whose acceleration and deceleration patterns make the use of internal combustion engines inefficient. Therefore, this kind of transport can be optimized using electric trucks [2]. Electric urban buses produce zero tailpipe emissions and reduce pollution and noise [3]. However, range is probably the major limiting factor for the mass use of electric vehicles in these cases. In order to improve the range of electric vehicles, the key points to be considered are energy consumption and vehicle performance.

According to the literature, numerous attempts have been made to develop different methodologies for optimizing the powertrain in electric vehicles [4], taking into account one or more of their components (battery pack, electric machine, transmission, electric devices for vehicles, and control system).

Different design variables have been analyzed in relation to the different purposes of the optimization of the powertrain. Lei et al. [5] considered power performance, energy consumption, and ride comfort using an in-wheel motor vehicle model as the basis for their multi-objective problem. They used two approaches to carry out the optimization, first a weighted objective method to transform the multi-objective optimization into a single-objective optimization, and a second approach to obtain the optimal solution from the Pareto front. In this paper, energy consumption was evaluated by the urban driving cycle (ECE-15).

Domingues-Olavarría et al. [6] also took into account the cost of the powertrain as an additional constraint. To do it they used a model, simulated over a given drive cycle, that takes into account the required size, performance, efficiency, and thermal characteristics of the main components in an electric powertrain.

Kulik et al. [7] estimated the requirements for a hybrid electric powertrain based on the analysis of a city vehicle GPS track together with accelerometer data. From the acceleration and velocity data of the track, the instantaneous power developed by the car is estimated and areas where regeneration is possible during braking.

Gearboxes are used for reasons of efficiency [8], in addition to improving the vehicle’s range and performance [9]. Different geared systems over the NEDC (New European Driving Cycle) were tested and the energy consumption was calculated.

Transmissions with a gear ratio (i.e., a gearbox) have been optimized by Dagci et al. [10], among others. Standard drive cycles are used.

Guo et al. [11] proposed a method for controlling the gearshift command in multispeed EV, reaching a reduction of 3–5% in energy consumption for city driving cycles. They used an algorithm, combining Pontryagin’s minimum principle and a numerical method, bisection method, for determining the gear positions and gearshift points. The algorithm was tested using the New European drive cycle (NEDC) and Urban Dynamometer Driving Schedule (UDDS).

Yu et al. [12,13] minimized the time consumed by a vehicle to travel along a given route or track, both for two independent wheel driving and four independent wheel driving. Describing a very detailed vehicle model, they solve a large-scale nonlinear optimization problem constrained to a certain track. The base speed, the constant power speed ratio, the static braking force distribution and the gear ratio were selected as the design parameters of the vehicle. The methodology does not optimize the vehicle mass and do not consider the energy consumption.

Xin [14] developed an optimal design of the powertrain system in order to minimize the curb mass of the vehicle, due to the sensitivity of vehicle mass for energy consumption. To carry out the simulation the Urban Driving Cycle (UDC), the NEDC cycle, and a constant speed cycle were chosen.

However, powertrain optimization is not the only way to improve the range of a vehicle. In order to improve the driving range of electric vehicles, especially for fleets of electric commercial vehicles, many studies [15] focus their efforts on planning the most efficient route. These studies use an average driver and are programmed to follow a driving cycle that include speed variation, topography, time stopped, and changes in payload. They are focused on the track, not on the vehicle or the travel time.

Other authors [16] focused their research on defining the best acceleration profile so that the energy consumed is minimized. Both a constant value of acceleration and a multi-step acceleration approach are considered in this last reference.

Energy consumption is a recurrent issue in this field, although battery life is not always regarded as an important variable. Liu et al. [17] optimized an acceleration profile for an electric vehicle, taking into account the energy consumption and percentage of battery capacity loss per kilometer. Yu et al. [18] also considered the battery life in the optimization process of an electric vehicle. In their paper, the sizing and energy management of the battery pack arranged in a hybrid racing vehicle is addressed as a multi-objective optimization problem.

Multi-objective optimization algorithms have also been applied to the design of complex mechanical systems [19] and, in particular, to the design of the powertrain of a hybrid vehicle [20]. Among multi-objective algorithms, evolutionary algorithms are widely used methods because of their flexibility and ease of use. Depending on the number of objectives, many different types of multi-objective evolutionary algorithms (MOEAs) exist. For problems with a moderate number of objectives, Pareto optimization algorithms such as the well-known NSGA2 (Non-dominated Sorting Genetic Algorithm–2) [21] and SPEA2 (Strength Pareto Evolutionary Algorithm–2) [22], among others, are effective and allow a set of solutions to be obtained in a single run (see [23] for a recent review of many-objective optimization algorithms). There are also scalarization strategies that transform the multi-objective problem into a scalar-valued optimization problem [24]. These last techniques are either used to rank incomparable solutions in the optimization algorithm or to select the most suitable candidate from the set of solutions produced by the MOEA.

Evolutionary multi-objective algorithms have also been used in problems related to the case at hand, such as the definition of operational strategies. For instance, MOEAs were used in [25] to reduce the operational costs of fleets of hybrid electric trucks.

As can be seen, many research methods have been applied to optimize the powertrain of a specific vehicle and were validated according to a standard driving test cycle [26]. Therefore, the need to develop an optimization method that defines the optimal chassis design for an electric vehicle, not only for the powertrain system, but also for a specific route is evident.

The methodology proposed in this paper is intended not only to optimize the powertrain, but also to minimize energy consumption by determining the optimal chassis design for a given track, setting the travel time as an objective. This enables adapting the chassis design to the path selected by the driver, being applicable to transport vehicles that cover one or more fixed routes. On the other hand, a standard driving cycle is not used. A driving mode is defined that fully approves the performance of the vehicle based on the set path. This allows for the optimization of acceleration and braking cycles and speeds to minimize energy consumption.

In addition to other resistance forces normally considered, the vehicle model takes into account the effects of curves in order to improve the accuracy of the evaluation of energy consumption [27].

A previously set path and time allows the designer to find a solution adapted to the specific needs of the vehicle. The use of multi-objective genetic algorithms to solve this issue will serve as a reliable generic method for finding the optimal configuration of the chassis on any given route.

As a particular case, it will allow the design of gearboxes with an optimal relation for each route or the definition of an automatic change strategy depending on the chosen route.

The case study in this paper is based on an electric race car, in which the values of the track and time were obtained from previous tests.

## 2. Materials and Methods

### 2.1. Optimization Methodology

An optimization methodology has been developed to achieve an optimal solution to the problem. The objective is to minimize the consumed energy with a maximum (reference) time on a given track. The starting information is as follows:With regard to the vehicle chassis. Each of these parameters can be assumed to be given data (fixed parameter) or may be optimized (design variables):○Geometrical and inertial values, such as vehicle mass, $m_{v}$, radius of the wheels, $r_{w}$, wheelbase, L, and position of the center of gravity, $l_{1}$, as shown in Figure 2.○Powertrain characteristics: related to the electrical motor (defined by its maximum power, $p_{max}$, maximum torque, $T_{max},\;$and maximum rotation speed, $n_{max}$. It is assumed that the engine always runs at maximum performance), mass of the battery pack, $m_{b},$ and transmission ratio, $i_{t}$.With regard to the track/road. The track is divided into several sections with constant values of different parameters. The information related to the track and related to each section consists of the:○Number of sections.○Initial and final point measured from the start point.○Radius of the curves.○Slope.○Grip or friction coefficient (longitudinal, $\mu_{x}$, and lateral, $\mu_{y}$).○Posted speed.Related to the optimization case:○Reference time, $t_{ref}$.Particular service constraints (limit accelerations due to passenger comfort or cargo safety, for instance).

The data related to the track act as constraints of the problem and need to be obtained prior to the optimization of the chassis. Additional information, such as initial or final speeds, stops or delays can be included in the numerical definition of the problem.

As to the subject of this paper, a multi-objective genetic algorithm is put forward to optimize the chassis of any kind of electric vehicle that performs a pre-established, known route, such as that shown in Figure 1.

This algorithm will generate the initial values of the design variables, which will be introduced in the calculation sequence (Simulator), obtaining a value of the error with respect to the references (time, minimum of energy, etc.). Due to its genetic character, it will take the best values of each generation as a reference to create the initial values of the next generation.

The process starts with the definition of the boundaries of the problem. This will give the algorithm a set of points within which to find the solution. In this part of the process, some settings of the algorithm are established such as the number of iterations, the tolerance for the error, and so on.

Each specimen in the population represents a set of designs variables of the chassis. The simulator evaluates this configuration of chassis and calculates the final time ($t_{final}$) and the consumed energy ($e_{c}$). These interim results will pass to an error function, where they will be compared with a reference time ($t_{ref}$) and the stored energy $\left(e_{b}\right)$ in the battery pack.

The final time obtained in the simulation, $t_{final}$, is related to the reference time set as an objective for the vehicle. This reference time is subtracted from the result time.
(1)$$\varepsilon_{t}=\left|t_{ref}-t_{final}\right|$$
where $\varepsilon_{t}$ is the error of the final time.

The consumed energy, $e_{c},$ is compared with the available stored energy in the battery pack, $e_{b}$, to check whether there is enough energy for the track. The stored energy in the battery pack for a certain mass is calculated as follows:(2)$$e_{b}=m_{b}\cdot \beta$$
where $e_{b}$ is the total stored energy in the battery pack and β is the energy density of the battery pack. The energy error, $\varepsilon_{e},$ is calculated as:(3)$$\varepsilon_{e}=\left|e_{c}-e_{b}\right|$$

The objective of the methodology is to minimize the overall error, E. This overall total error is calculated as a weighted sum of the two aforementioned errors and the consumed energy with its own weighting$\;(k_{c})$:(4)$$E=k_{t}\cdot \varepsilon_{t}+k_{e}\cdot \varepsilon_{e}+k_{c}\cdot e_{c}$$
where $k_{t}$ is the value of the weighting associated with the time error and $k_{e}$ is the error associated with the energy. Once the total error has been calculated for every specimen in the population, the best values are selected and crossed over to define the next generation.

If the stopping criteria are reached, the best values of the last generation will be the optimization result. The objective of the methodology is to obtain a set of non-dominated designs with respect to the three aforementioned criteria: $\varepsilon_{t}$, $\varepsilon_{e}$ and $e_{c}$. The NSGA2 algorithm [21] is used to solve this vector-valued optimization problem. A scalarization strategy [24] is used to select a suitable candidate from the final set of Pareto-optimal solutions produced by the MOEA.

### 2.2. Mathematical Model of Longitudinal Dynamics

The definition of a mathematical model of the behavior of the vehicle is subject to the following first-order dynamic constraints:(5)$$\dot{x}\left(t\right)=f\left[x\left(t\right),u\left(t\right),t,p\right]$$
where $\dot{x}$ is the first-order derivative of the state variables, f is the dynamic model, and x, u, p are, respectively, the state, input, and design vectors.

The state vector, x**,** includes the variables used to describe the speed, v(t), and position, x(t), from the start point of the track:(6)x(t)=[v(t),x(t)]

The input vector includes the longitudinal forces acting on the vehicle:(7)$$u=\left[R_{r},R_{g},R_{a},R_{c},F\right]$$
where the rolling resistance, $\left(R_{r}\right)$, grade resistance, $\left(R_{g}\right)$, aerodynamic drag, $\left(R_{a}\right)$, curve resistance, $\left(R_{c}\right)$ [27], and traction/braking force, (F) are considered. The values and equations are discussed later.

The considered design variables are related to the design of the chassis, including geometrical, inertial and powertrain characteristics. These variables are included in the design vector, (p), as follows:(8)$$p=\left[m_{v},r_{w},\;L,\;l_{1},p_{max},T_{max},n_{max},\;m_{b},i_{t}\right]$$

The derivation of the first-order dynamic constraints is proposed according to Newton’s Second Law [28], in which: “The sum of the external forces acting on a body in a given direction is equal to the product of its mass and the acceleration in that direction (assuming the mass is fixed)”:(9)$$\sum F_{ext}=m_{t}\cdot \frac{dv\left(t\right)}{dt}$$
where $F_{ext}$ represents the external forces and $m_{t},$ the total mass of the vehicle.

The external forces acting on a two-axle vehicle [29] are shown in Figure 2. These include:

$R_{a}$: Aerodynamic resistance:(10)$$R_{a}=\frac{1}{2}\rho C_{x}A_{f}{\left(v\left(t\right)\right)}^{2}$$

$R_{rf},R_{rr}$: Rolling resistance of the front and rear tires, respectively, where $f_{r}$ represents the rolling resistance coefficient and $m_{t}=m_{v}+m_{b}$:(11)$$R_{r}=R_{rf}+R_{rr}=m_{t}\;g\cdot f_{r}$$

$R_{g}$: Grade resistance:(12)$$R_{g}=m_{t}\;g\cdot \sin\theta$$

The value of the ramp will be determined at each time instant depending on the position of the vehicle on the road under study: θ=θ(x(t)).

$F_{f},F_{r}$: Tractive or braking effort of the front and rear tires, respectively:(13)$$F\left(t\right)=F_{f}+F_{r}$$

The value of the force, F(t), will be discussed later. The first-order dynamic constraints (5), including all the forces applied to the vehicle, can be expressed as follows:(14)$$\begin{array}{c} \frac{dv\left(t\right)}{dt}=\frac{F\left(t\right)-\left(R_{c}\left(v\left(t\right)\right)+m_{t}\;g\cdot f_{r}+m_{t}\;g\cdot \sin\theta \left(x\left(t\right)\right)+\frac{1}{2}\rho C_{x}A_{f}{\left(v\left(t\right)\right)}^{2}\right)}{m_{t}\gamma }\\ \frac{dx\left(t\right)}{dt}=v\left(t\right) \end{array}]\forall t/0\leq x\left(t\right)\leq x_{finish}$$
where g is the gravitational acceleration, ρ is the density of the air, $C_{x}$ is the aerodynamic coefficient of the vehicle and $A_{f}$ is the frontal area of the vehicle. A mass factor, γ, is introduced to take into account the effect of the inertia of the rotating parts.

To predict how the vehicle will behave on the route, the first-order dynamic constraints will be integrated in a given path for every set of values of the design variables. The above differential equations system is integrated depending on the road and the vehicle characteristics so as to determine its position, x(t), and speed, v(t).

The force, F(t), should be calculated according to numerous variables as discussed later.
(15)$$F\left(t\right)=F\left(x\left(t\right),v\left(t\right),\mu_{x},\mu_{y},r_{w},\;i_{t},\;T_{max},p_{max},n_{max,\;}\;L,\;l_{1},h,m_{t},\eta_{b},a_{x}\right)$$

On each section of the track, the force, F(t), may be tractive or braking, as will be seen in Section 2.3 Track Characterization. This is linked to the possibility of the vehicle accelerating, maintaining a constant speed or braking to a certain final speed.

The two factors that limit the maximum tractive effort are $F_{T\_\mu }$ and $F_{T\_m}$. $F_{T\_\mu }$ is related to the friction coefficient, $\mu_{x}$, and the normal load on the driven axle. $F_{T\_m}$ is related to the characteristics of the vehicle’s powertrain:(16)$$F_{T}=\min\left\{F_{T\_m},F_{T\_\mu }\right\}$$

In some cases, however, the tractive effort could result in unsuitable acceleration for passenger comfort or cargo safety. Hence, depending on the case under study, a maximum longitudinal acceleration, $a_{x},$ should be established that is compared with the value obtained in Equation (14). The minimum value is introduced in the first-order dynamic constraint (5) and the demanded tractive effort is calculated.

On the other hand, it is necessary to check whether the vehicle passes the speed limit in each section.

In order to perform this calculation, the first-order dynamic constraints are integrated from the beginning to the end of the track. The position of the vehicle with respect to the road is obtained via numerical integration to identify which section the vehicle is in at any instant in time.

The whole track is simulated, obtaining the overall time and overall energy consumed by the vehicle. The evaluation of the tractive or braking force is discussed in the following subsection.

#### 2.2.1. Tractive Effort due to the Powertrain

The electric motor is responsible for converting electrical energy into mechanical energy, consuming the stored power in the battery in order to move the transmission set. The behavior (Figure 3) of the motor defines the speed, torque and other important variables for the performance of the vehicle.

To mathematically define the powertrain of an electric vehicle, the starting point is the maximum torque and power values of the motor. The motor power, p, is defined as:(17)$$p=T\cdot n_{m}$$
where T is the motor torque and $n_{m}$, the rotational speed. The changeover point between maximum torque and constant power (cut-off speed, $n_{m\_c}$) is defined as:(18)$$p_{max}=T_{max}\cdot n_{m\_c}$$
and can be calculated via the following expression:(19)$$n_{m\_c}=\frac{p_{max}}{T_{max}}$$

Thus, the characteristic torque versus speed curve for the motor can be defined as:(20)$$T\left(n_{m}\right)=\{\begin{array}{c} T_{max},\;\;n_{m}<n_{m\_c}\\ \frac{p_{max}}{n_{m}},\;\;n_{m}\geq n_{m\_c} \end{array}$$

Knowing the rotational speed of the motor, it is possible to calculate the vehicle’s longitudinal speed (v):(21)$$v=\frac{n_{m}}{i_{t}}\cdot r_{w}$$
where $r_{w}$ is the effective radius of the wheels on the drive axles. The slip phenomena are neglected in this model. The top speed $\left(v_{top}\right)$ due to the powertrain can be calculated via the following expression:(22)$$v_{top}=\frac{n_{max}}{i_{t}}\cdot r_{w}$$

The torque applied at the wheels, $T_{w}$, by the powertrain can be calculated as follows:(23)$$T_{w}=T\cdot i_{t}$$

The tractive force of the motor due to wheel torque, $F_{T\_m}$, is defined as:(24)$$F_{T\_m}=\frac{T_{w}}{r_{w}}$$

This force will be constant, until the vehicle reaches the speed at which the torque stops being constant, which is called the cut-off speed or $v_{c}$. This speed is related to the variables of the motor via the expression:(25)$$v_{c}=\frac{n_{m\_c}}{i_{t}}\cdot r_{w}=\frac{r_{w}}{i_{t}}\cdot \frac{p_{max}}{T_{max}}$$

Bearing this last equation in mind, a new traction force equation can be described in terms of the cut-off speed. This can be expressed as:(26)$$F_{T\_m}\left(v\right)=\{\begin{array}{c} T_{max}\cdot \frac{i_{t}}{r_{w}},\;\;v<v_{c}\\ \frac{p_{max}}{n_{m}}\cdot \frac{i_{t}}{r_{w}}=\frac{p_{max}}{v},\;\;v\geq v_{c} \end{array}$$

These are the maximum tractive forces related to the powertrain, which will later be compared with the maximum efforts due to the grip of the track.

Another element to consider in order to obtain a reliable model of the vehicle is the battery. The existence of efficiencies in the transfer of power and its subsequent use by the motor should be taken into account. These efficiencies have to be considered so that the necessary power is not underestimated when modeling the vehicle.

Assuming $p_{b}$ to be the total demanded power needed to drive the vehicle and $p_{m}\;$to be the output power of the battery, then its efficiency, $\eta_{bat},$ is defined as:(27)$$\eta_{bat}=\frac{p_{m}}{p_{b}}$$

Furthermore, $p_{m}$ is usually modeled as a function of the vehicle’s mechanical power and the machine or motor efficiency, $\eta_{m}$:(28)$$p_{m}=\frac{T\cdot \;n_{m}}{\eta_{m}\left(T,\;n_{m}\right)}$$

As can be seen, the motor efficiency is a function of the speed and torque output multiplied by itself. This efficiency normally reaches its maximum in lower speed regions compared to an internal combustion engine [11]. A polynomic expression can be used to obtain a description of the behavior of the power demanded by the vehicle. A two-dimensional polynomial expression can be used to obtain this value, as follows:(29)$$p_{b}=\frac{T\cdot n_{m}}{\eta_{m}\cdot \eta_{bat}}\approx \overset{2}{\underset{i=0}{\sum }}\overset{2}{\underset{j=0}{\sum }}p_{ij}T^{i}\;n_{m}{}^{j}$$
where $p_{ij}$ are the fitting coefficients or tuning values. These values are set to effectively represent the motor power [30]. The value of these fitting coefficients can be obtained by running a set of tests. It is easier to work with this approximate closed-form (29) expression, instead of using the analytic version (28).

#### 2.2.2. Tractive Effort due to the Grip of the Track

In order to define the maximum tractive effort due to the grip in a rear/forward wheel-drive vehicle, the first step is to determine the normal loads on the vehicle’s axles. The normal load on the rear axle is related to the sum of the moments acting on that axle:(30)$$W_{r}=\frac{m_{t}\;gl_{1}\cos\theta +R_{a}h_{a}+h\;am_{t}+m_{t}\;gh\sin\theta \;}{L}$$
where $l_{1}$ is the distance between the front axle and the vehicle’s center of gravity and, in this equation, the sign of θ is positive when the vehicle is climbing a hill.

The normal load on the front axle is related to the sum of the moments acting on that axle:(31)$$W_{f}=\frac{m_{t}\;gl_{2}\cos\theta -R_{a}h_{a}-h\;am_{t}+m_{t}\;gh\sin\theta \;}{L}$$
where $l_{2}$ is distance from the rear axle to the vehicle’s center of gravity and, in this equation, the sign of θ is negative when the vehicle is climbing a hill. Assuming that the point of application of the aerodynamic resistance ($h_{a}$) is near the height of the center of gravity (h), the last two equations may be rewritten as:(32)$$W_{f}=\frac{l_{2}}{L}m_{t}\;g-\frac{h}{L}\left(R_{a}+am_{t}+R_{d}+\sin\theta \left(x\left(t\right)\right)\right)$$
and:(33)$$W_{r}=\frac{l_{1}}{L}m_{t}\;g-\frac{h}{L}\left(R_{a}+am_{t}+R_{d}+\sin\theta \left(x\left(t\right)\right)\right)$$

Substituting Equation (11) in the last equation, we have that:(34)$$W_{r}=\frac{l_{1}}{L}m_{t}\;g+\frac{h}{L}\left(F-R_{r}\right)$$
and:(35)$$W_{f}=\frac{l_{2}}{L}m_{t}\;g-\frac{h}{L}\left(F-R_{r}\right)$$

Using this last equation, the maximum tractive force of the vehicle can be determined for a rear wheel-drive vehicle as:(36)$$F_{T\_\mu \_r}=\mu_{x}W_{r}=\mu_{x}\left[\frac{l_{1}}{L}m_{t}\;g+\frac{h}{L}\left(F_{max}-R_{r}\right)\right]$$
and:(37)$$F_{T\_\mu \_r}=\frac{\mu_{x}W\left(l_{1}-f_{r}h\right)/L}{1-\mu h/L}$$

For a front wheel-drive vehicle, the maximum tractive force is:(38)$$F_{T\_\mu \_f}=\mu_{x}W_{f}=\mu_{x}\left[\frac{l_{2}}{L}m_{t}\;g+\frac{h}{L}\left(F_{max}-R_{r}\right)\right]$$
and:(39)$$F_{T\_\mu \_f}=\frac{\mu_{x}W\left(l_{2}+f_{r}h\right)/L}{1-\mu h/L}$$

#### 2.2.3. Braking Dynamics

While studying the behavior of the vehicle on each section of the route, when the speed is higher than the final speed of the section, it will be necessary to brake in order to get to the end of the section at the specified speed. No regenerative braking is considered. To do so, it is necessary to determine the braking point. This point is related to the braking performances of the vehicle. These performances are controlled by the braking force. To determine this force, the values of the maximum grip force in the longitudinal direction, $\mu_{x},$ and the braking efficiency, $\eta_{b},$ need to be obtained. The braking force of a vehicle can be determined as:(40)$$F_{b}=m_{t}g\mu_{x}\cdot \eta_{b}$$

The braking acceleration can be calculated as:(41)$$a_{b\_\mu_{x}}\left(t\right)=\frac{-F_{b}-\left(m_{t}\;g\cdot f+m_{t}\;g\cdot \sin\theta +\frac{1}{2}\rho C_{x}A_{f}{\left(v\left(t\right)\right)}^{2}\right)}{m_{t}\gamma }$$

In some cases, however, the braking acceleration (41) might not be suitable for passenger comfort or cargo safety. Hence, depending on the case, a maximum longitudinal acceleration $\left(a_{x}\right)$ should be established that is compared with the value obtained in Equation (41). The minimum value is consequently chosen:(42)$$\left|a_{b}\left(t\right)\right|=\min\left\{\left|a_{b\_\mu_{x}}\left(t\right)\right|,\left|a_{x}\right|\right\}$$

Considering the braking acceleration as constant in every section, the braking distance, $d_{b},$ to decrease the actual speed, v(t), to a final speed, $v_{final},$ can be calculated as follows:(43)$$d_{b}\left(v\left(t\right)\right)=\frac{v_{final}{}^{2}-v{\left(t\right)}^{2}}{a_{b}}$$

### 2.3. Track Characterization

A road route can have a complex three-dimensional geometry, with straight and curved alignments as well as elevation. It is therefore necessary to apply the dynamics equations to the entire route.

In the proposed methodology, the Geographic Information System (GIS) data is pre-processed before modeling. The road centerline (Figure 4) is determined by analyzing the GIS data and orthophotos. Subsequently, the precision of the route is improved and adapted to the terrain by combining the horizontal alignment and elevation profile.

The geometrical data on the route is exported and divided into parametrized sections with their corresponding length, curvature slope, and grip values.

Following this analysis of the track, the result thus obtained is a set of (n) sections. Each section (i) is characterized by several parameters:Longitudinal coordinate of the initial, $x_{ini}\left(i\right),$ and final, $x_{final}\left(i\right),$ points.Elevation above sea level of the initial, $z_{ini}\left(i\right),$ and final, $z_{final}\left(i\right),$ points.Radius of curvature, $r_{c}\left(i\right)$.Section grip ($\mu_{x}\left(i\right)$, $\mu_{y}\left(i\right)),$ which is characterized by the vehicle dynamics limit. This parameter may be calculated or obtained from an instrumented vehicle.

The final point of a section (*i*) matches the initial point of the next one (*i* + 1). This means that each section starts right where the previous one ends; that is:(44)$$x_{ini}\left(i+1\right)=x_{final}\left(i\right)$$

It will be assumed that the start of the track has a value of $x_{ini}\left(1\right)=0$ and the finish, a value of $x_{ini}\left(n+1\right)=x_{finish}=$ length of the track.

This allows the length of each section, l(i), to be characterized:(45)$$l\left(i\right)=x_{final}\left(i\right)-x_{ini}\left(i\right)$$

Hence, when the vehicle, at time step t, is located on section i, the following holds:(46)$$x_{ini}\left(i\right)\leq x\left(t\right)\leq x_{final}\left(i\right)$$

The slope of each section, θ(i), which is assumed to be constant, can be calculated as:(47)$$\theta \left(i\right)=\frac{z_{final}\left(i\right)-z_{ini}\left(i\right)}{x_{final}\left(i\right)-x_{ini}\left(i\right)}\cdot 100$$

Each section is characterized by a maximum speed $\left(v_{max}\left(i\right)\right)$ that is related to the vehicle and the track. The vehicle presents a top speed, $v_{top},$ related to the maximal rotational speed of the motor. For each section i, the track itself imposes another speed limit, $v_{a_{y}}\left(i\right)$. This value is related to the maximum lateral acceleration due to particular limitations (passenger comfort, cargo safety, etc.), $a_{y\_pc}\left(i\right),$ and the lateral friction limit, $\mu_{y}\left(i\right)$. The maximum speed, $v_{a_{y}}\left(i\right),$ of section i is calculated as follows:(48)$$v_{a_{y}}\left(i\right)=\min\left\{\sqrt{a_{y\_pc}r_{c}\left(i\right)},\sqrt{\mu_{y}\left(i\right)gr_{c}\left(i\right)}\right\}$$

An additional limitation may appear, depending on traffic rule constraints on each section of the track, denoted as $v_{posted}\left(i\right)$. The maximum speed, $v_{max}\left(i\right),$ of section i is the minimum of these three values:(49)$$v_{max}\left(i\right)=\min\left\{v_{top},v_{a_{y}}\left(i\right),v_{posted}\left(i\right)\right\}$$

Once the sections of the track have been defined, the behavior of the vehicle as it passes through each one needs to be analyzed.

### 2.4. Driving Dynamics and Energy

The proposed algorithm takes the initial speed of a section (i) as the final speed of the immediately previous one (i − 1).

As a rule of thumb, when the maximum value among the tractive forces is the one related to the powertrain, the powertrain will be used to the maximum, but the maximum speed, $v_{max}\left(i\right),$ can never be passed. Moreover, if the vehicle is passing through a section (i) at a higher speed than the predefined final speed, $v_{final}\left(i\right),$ it needs to brake.

To understand the vehicle response in each section, a series of rules must be considered:The initial speed of section i is known, $v_{ini}\left(i\right)$, determined by the algorithm as the final speed of the previous section (i−1).If the speed is below the maximum for this section and there is no need to brake, the case will be ‘Traction’.
(50)$$if\;\left[v\left(t\right)<v_{max}\left(i\right)\;\wedge not\left[case:\mathrm{BRAKING}\right]\right]\;\;then{\;}^{\prime }case:{\mathrm{TRACTION}}^{\prime }$$At the end of a section (i), the speed cannot be greater than the maximum speed, for the next section$,\;v_{max}\left(i+1\right)$.
(51)$$if\;\left[x\left(t\right)=x_{ini}\left(i+1\right)\right]\;then\;v\left(t\right)\leq \;v_{max}\left(i+1\right)$$

The finish speed of a section, $v_{final}\left(i\right)$, cannot be higher than the speed limit for the next section, $v_{lim}\left(i+1\right)$.


If the speed $v\left(t\right)>v_{max}\left(i+1\right)$, the vehicle must brake. The braking process must ensure that the final speed at the end of the section (*i*), is also the maximum speed for the next section (*i* + 1), namely:(52)$$v_{final}\left(i\right)=v_{max}\left(i+1\right)$$


Considering that the braking acceleration, $a_{b}\left(i\right)$, is constant in the section, the braking distance, $d_{b},$ can be calculated as follows:(53)$$d_{b}\left(i,v\left(t\right)\right)=\frac{v_{final}{\left(i\right)}^{2}-v{\left(t\right)}^{2}}{a_{b}\left(i\right)}$$

The calculation algorithm must check, during each time step, the distance between the end of the section and the current position of the vehicle in order to activate the braking case, or not:(54)$$if\;\left[\left(x_{final}\left(i\right)-x\left(t\right)\right)\leq d_{b}\left(i,v\left(t\right)\right)\;\wedge v\left(t\right)>v_{final}\left(i\right)\right]\;then{\;}^{\prime }case:={\mathrm{BRAKING}}^{\prime }$$

In the case of maximal longitudinal acceleration, as shown in Figure 5, the drive through a section can be defined via four different dynamic scenarios:The vehicle accelerates from $v_{ini}\left(i\right)$ to $v_{c}$ at constant torque. The total force in this subsection is:(55)$$if\;\left[case=\mathrm{TRACTION}\;\wedge \left(v\left(t\right)<v_{c}\right)\right]\;\;then\;F=\frac{T_{max}i_{t}}{R}$$The vehicle accelerates from $v_{c}$ to $v_{max}\left(i\right)$ at constant power. The total force in this subsection is:(56)$$if\;\left[case=\mathrm{TRACTION}\;\wedge \left(v_{c}<v\left(t\right)<v_{max}\left(i\right)\right)\right]\;then\;F=\frac{p_{max}}{v}$$The vehicle circulates at constant speed, $v_{max}\left(i\right)$. In this case, the force will be that needed to overcome the forces of resistance.
(57)$$F=\left(m_{t}\;g\cdot f_{r}+m_{t}\;g\cdot \sin\theta \left(i\right)+\frac{1}{2}\rho C_{x}A_{f}{\left(v_{max}\left(i\right)\right)}^{2}\right)$$The vehicle brakes from $v_{max}\left(i\right)$ to $v_{f}$. The total force of this subsection is:(58)$$if\;\left[case=\mathrm{BRAKING}\right]\;then\;F=F_{b}$$

A way to calculate the energy consumed by the vehicle on its route is needed in order to achieve optimization. This energy can be calculated as:(59)$$e_{c}=\int_{0}^{t_{f}}p_{b}\left(t\right)dt$$
where $e_{c}$ is the overall consumed energy.

## 3. Results

A study case was devised to implement and validate the proposed methodology. The case is defined by the track and the vehicle. The vehicle used as a reference for the implementation of the algorithm is an ICE (internal combustion engine) rear-wheel drive race car (as shown in Figure 6), designed for hill climb competitions (Table 1). The hill climb discipline is perfect for the use of electric vehicles due to the track length [30].

The optimization target is to run the chosen race track in the same time (reference time) as the ICE vehicle, consuming the minimum amount of energy. To do so, an electric motor was previously selected (Table 2).

The energy density of the chosen batteries cells is β=128 Wh/kg. According to these conditions, the design variables can be expressed in this case as:(60)$$p=\left[475,\;0.264,\;2.5,\;1.431,\;120,000,\;350,\;12,000,\;m_{b},i_{t}\right],$$
where several variables are fixed and assumed as parameters. In this particular case, the mass of battery packs, $m_{b},$ and the transmission ratio, $i_{t},$ are the design variables to be optimized.

The chosen track is a scoring event for the European Hill Climb Championship. The track characterization is based on the three-dimensional analysis of a LIDAR dataset (2008–2015), reference ETRS89, from the Spanish National Geographic Institute (26,355,144 points have been processed). The density of points is 0.5 points/m^2^ and the altimetry precision is better than 20 cm RMSE(z) (root mean square error).

Istram^®^BIM software (Buhodra Ingeniería S.A., Llanera, Asturias, Spain) was used to import and process the LIDAR (light detection and ranging) data and for 3D characterization of the route geometry (Figure 7). The main horizontal route and its elevation were defined, based on terrain data, an orthophoto and Spanish road regulations IC-3.1.

The track, as shown in Figure 7, has a length of 5.260 km with a maximal gradient of 6.65% and an average gradient of 5.96%. The altitude above sea level is 46 m at the beginning and 359 m at the end, as shown in Table A1—Appendix A. During the optimization process, the algorithm decomposed this track into 107 sections.

Real data recorded in this event (telemetry of the ICE vehicle) was used in the optimization procedure. According to these data, the braking acceleration presents a minimum constant value of −13 m/s^2^, the reference time is 168.5 s and the initial speed is 0 km/h. There is no restriction for maximum longitudinal or lateral acceleration due to passenger comfort or cargo safety, so high values are defined (above 6 *g*).

The parameters of the genetic algorithm are as follows:Population:○Population Size: 30○Initial Range: [5–100, 1–10]Selection:○Selection Function: Tournament.○Tournament Size: 2.Crossover:○Crossover Fraction: 0.8○Crossover Function: Intermediate.Mutation:○Mutation Function: Constraint dependent.Migration:○Direction: Forward.○Fraction: 0.2○Interval: 20Multi-objective Options:○Pareto Fraction: 0.35Stopping Criteria:○Maximal Generation: 400○Stall Generation: 100○Function Tolerance: 1 × 10^-4^

The final values of the optimization after 102 generations are shown in Table 3.

A design methodology based on standard driving cycles, without taking full advantage of the powertrain’s performance, obtains mass values of the battery pack greater than 50 kg. If the methodology does not take into account the precise values of the route, but instead consider average values, the mass values of the battery pack would be greater than 42 kg. The implementation of methods of optimization of the consumed energy, without taking into account the multivariable analysis, does not allow obtaining values lower than 40 kg of the mass of the battery pack.

Figure 8 shows the speed profile of the model with the optimized powertrain generated by the simulator. The new profile meets the requirements imposed on the model in terms of both energy consumption and elapsed time.

Figure 9 shows the plot of acceleration and speed versus distance, on the first kilometer of the track. The different vehicle dynamic scenarios are also identified in the figure (1: traction–torque max; 2: traction–power max; 3: constant speed; and 4: braking).

## 4. Discussion and Conclusion

Most published methods optimize the powertrain of electric vehicles in standard driving tests. In this contribution, a general methodology has been proposed that allows fine-tuning the chassis of electrical vehicles in an arbitrary route or set of routes. An approximate mathematical model has been developed that takes into account many different parameters of both the vehicle and track, such as geometries, inertial values, forces, resistances, grip, and powertrain, among others. This set of information enables accurate evaluation of the consumed energy and at the same time is simple enough for being computationally efficient, thus, it can be embedded in the fitness function of a multi-objective genetic algorithm.

This methodology has been used and validated in the optimization of a race car. The chosen track was a scoring event for the European Hill Climb Championship. In this case, the design variables were the battery pack mass and the transmission ratio, both optimized to ensure optimal performance on the track. The multi-objective model was validated with the use of a real-life case that confirms its functionality and accuracy. The implementation of the proposed methodology allows to reduce the weight of the battery pack by up to 20%, compared to traditional design methods.

For a more robust design, in future work the mathematical model should be able to cope with uncertainties in the acquired data and the state of the vehicle. We intend to deploy algorithms for multi-criteria optimization under uncertainty and obtain solutions are that are insensitive to small variations in weight of the loaded vehicle or the capacity of the batteries, to name some examples.

## Figures and Tables

**Figure 1 sensors-20-03633-f001:**
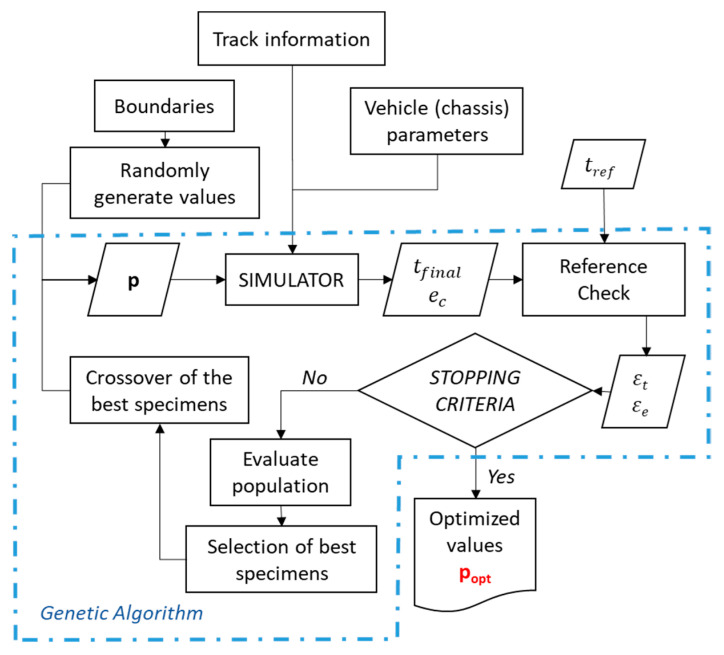
General flowchart of the proposed methodology.

**Figure 2 sensors-20-03633-f002:**
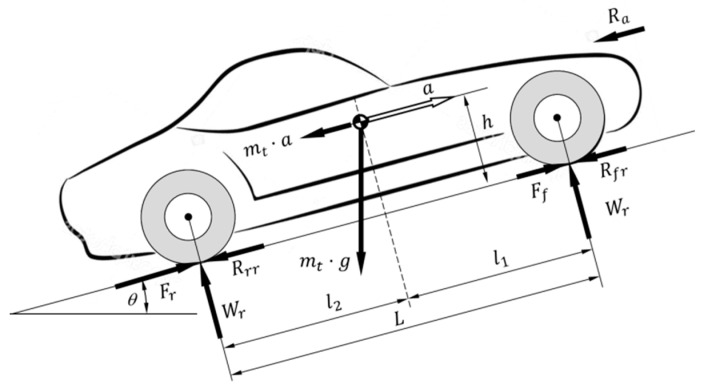
Forces acting on a vehicle.

**Figure 3 sensors-20-03633-f003:**
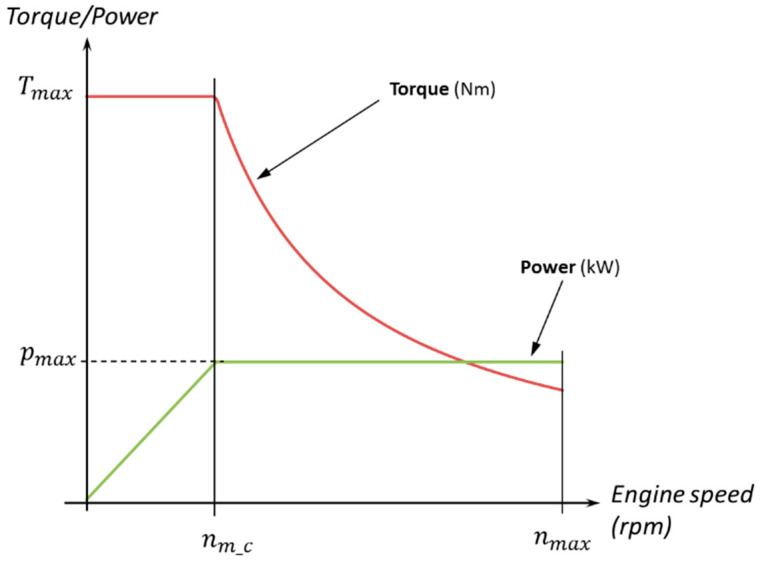
Torque and power curves of an electric motor.

**Figure 4 sensors-20-03633-f004:**
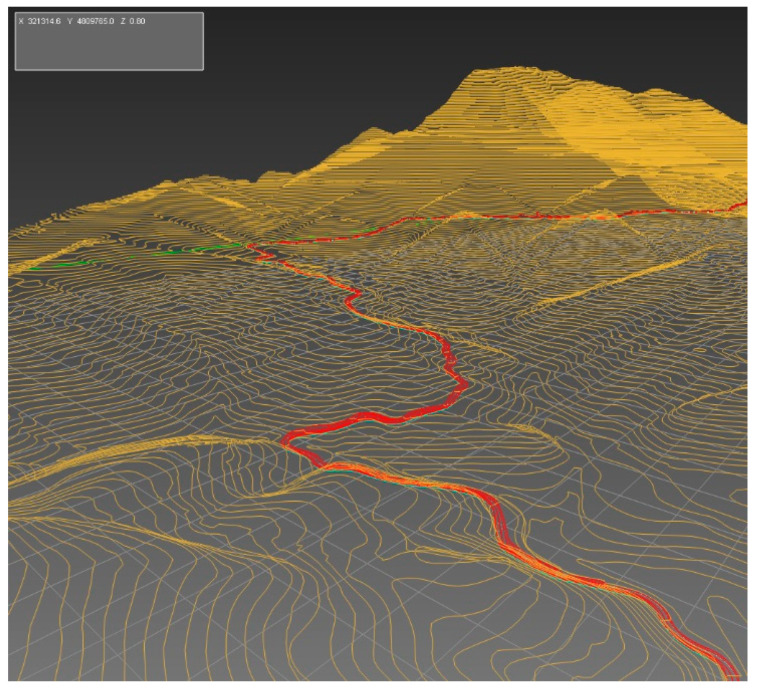
3D characterization of the track. The road centerline (red) is shown in a Geographic Information System using a digital model (yellow) of the study area and road.

**Figure 5 sensors-20-03633-f005:**
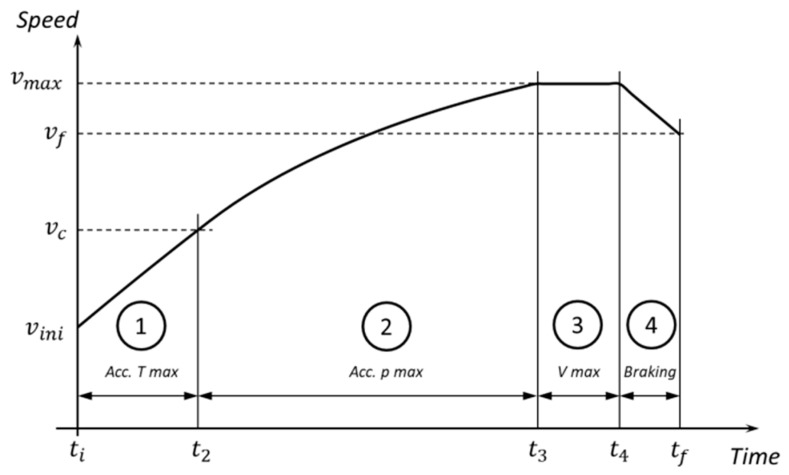
Vehicle dynamic scenarios on a section of the track.

**Figure 6 sensors-20-03633-f006:**
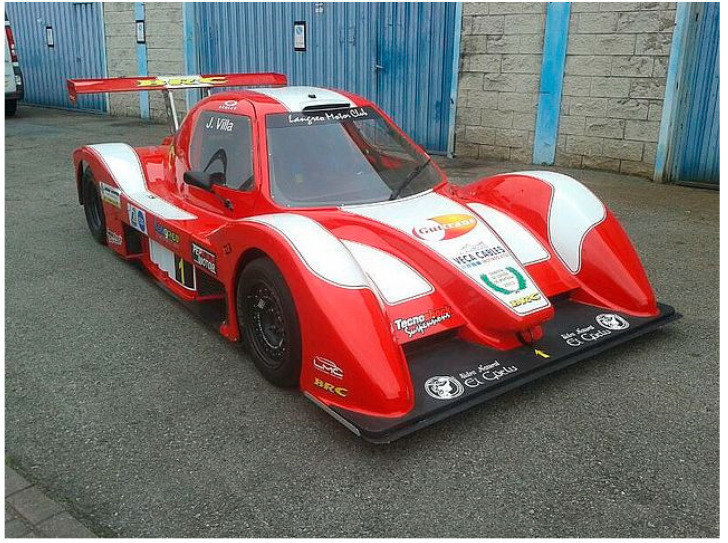
ICE vehicle.

**Figure 7 sensors-20-03633-f007:**
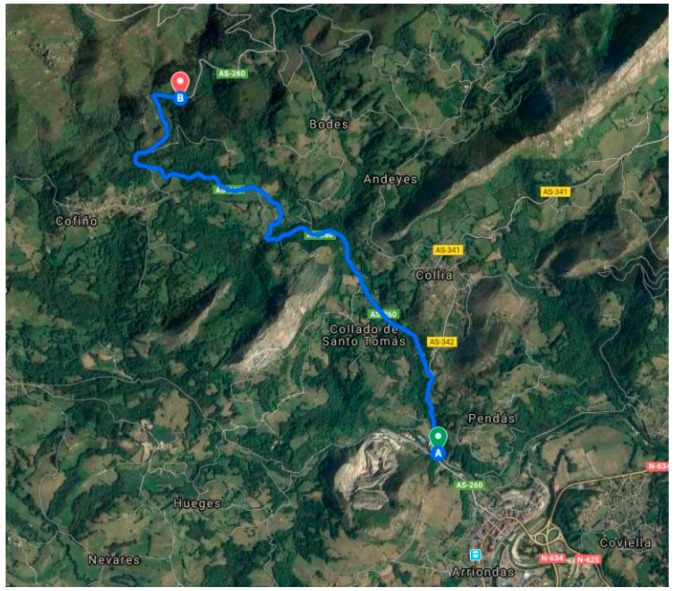
Map of the track.

**Figure 8 sensors-20-03633-f008:**
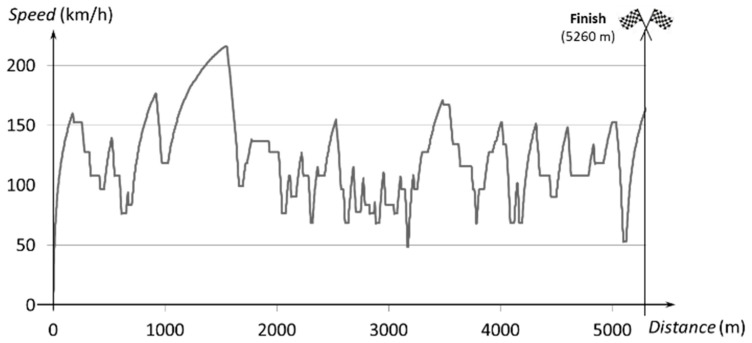
Speed vs. distance using the optimized chassis.

**Figure 9 sensors-20-03633-f009:**
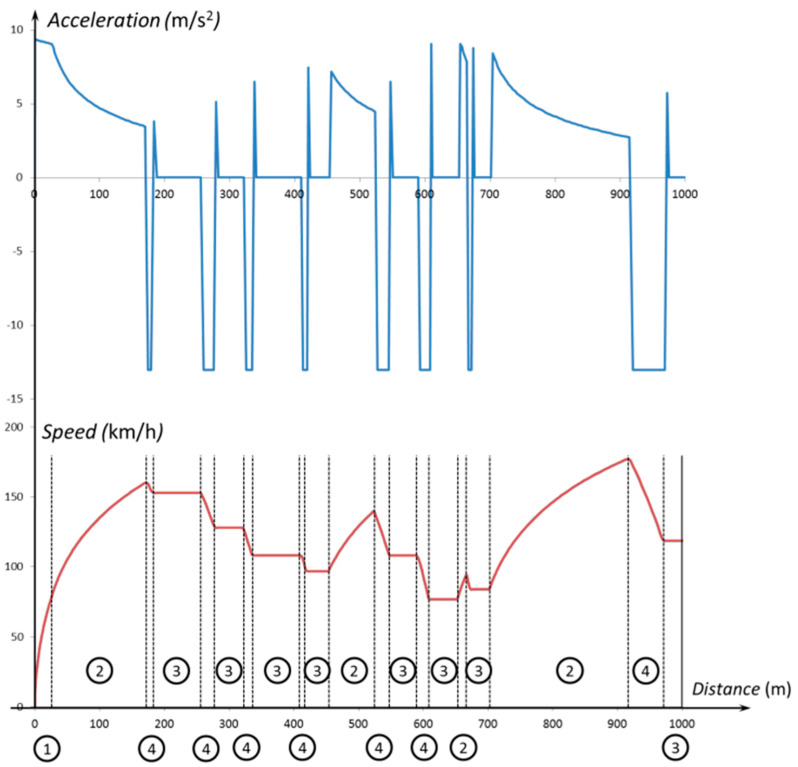
Acceleration and speed vs. distance on the first kilometer of track.

**Table 1 sensors-20-03633-t001:** Vehicle parameters.

$m_{v}\;\left(kg\right)$	Vehicle Mass without the Battery	475
L (m)	Wheelbase	2.5
r (m)	Wheel radius	0.264
$L_{v}\;\left(m\right)$	Length of the vehicle	3.750
$H_{v}\;\left(m\right)$	Height of the vehicle	1.030
$W_{v}\;\left(m\right)$	Width of the vehicle	1.750
$A_{f}\;\left(m^{2}\right)$	Frontal area of the vehicle	1.210
$C_{x}$	Aerodynamic coefficient	0.3

**Table 2 sensors-20-03633-t002:** Motor parameters.

$p_{max}\;\left(kW\right)\;$	Motor Maximum Power	120
Tmax (Nm)	Motor maximum torque	350
$n_{m}\;\left(rpm\right)$	Motor maximum rotational speed	12,000

**Table 3 sensors-20-03633-t003:** Optimization solution.

$t_{final}\;\left(s\right)\;$	Total Time	168.49999999999471
$m_{b}\;\left(\mathrm{kg}\right)$	Battery pack mass	35.12
$i_{t}$	Transmission ratio	4.011
$e_{c}\;\left(\mathrm{kWh}\right)\;$	Consumed energy	2.73

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
