# Peer review of "Multi-Objective Evolutionary Design of an Electric Vehicle Chassis"

_sensors, 2020, doi:10.3390/s20133633_

Round 1
Reviewer 1 Report
The study represents an interesting methodology to obtain the preferred design of an electric vehicle chassis. The manuscript can be improved by considering the following:
- In Abstract, please provide a summary of actual results obtained from the study in a quantitative way showcasing the advantages of the proposed methodology.
- Line 15, please remove ‘…’ and rephrase the sentence
- The literature review is thorough and well written. However, a few points are to be addressed:
- Authors have mentioned several studies on powertrain optimisation approaches adapted by researchers in between lines 31 – 51. Many of them just mentioned as what they have considered without explaining the detailed outcome and shortcoming of their methodology. Please revise this section.
- Please mention the full form of any acronym used for the first time, e.g. NSGA2 or SPEA2.
- Please summarise the current limitations within the literature and need for the present work to address those limitations.
- In the case study, the battery pack weight and the transmission ratio were optimised as vehicle chassis parameters. Please elaborate on the results obtained using your methodology with an existing method to demonstrate the usefulness of this method.
- Please elaborate the section 3 as Results and discussion. And then add a Conclusion section with future work.
Author Response
Thank you in advance for your careful review of this paper. In this letter, we have reproduced your comments and have followed each comment with our response.
- In Abstract, please provide a summary of actual results obtained from the study in a quantitative way showcasing the advantages of the proposed methodology.
The new version of the manuscript includes the following sentence:
The implementation of the proposed methodology allows to reduce the weight of the battery pack by up to 20%, compared to traditional design methods.
- Line 15, please remove ‘…’ and rephrase the sentence
The new version of the manuscript includes the following sentence:
“Every year, the European Union, USA, Japan, among others, set stricter targets for CO2 emissions for passenger cars and commercial vehicles.”
- The literature review is thorough and well written. However, a few points are to be addressed:
- Authors have mentioned several studies on powertrain optimisation approaches adapted by researchers in between lines 31 – 51. Many of them just mentioned as what they have considered without explaining the detailed outcome and shortcoming of their methodology. Please revise this section.
The new version of the manuscript includes the following sentences:
Different design variables have been analyzed in relation to the different purposes of the optimization of the powertrain. Lei et al. [5] consider power performance, energy consumption and ride comfort using an in-wheel motor vehicle model as the basis for their multi-objective problem. They used two approaches to carry out the optimization, first a weighted objective method to transform the multi-objective optimization into a single-objective optimization and a second approach to obtain the optimal solution from the Pareto front. In this paper, energy consumption was evaluated by the urban driving cycle (ECE-15).
Domingues-Olavarría et al. [6] also take into account the cost of the powertrain as an additional constraint. To do it they used a model, simulated over a given drive cycle, that takes into account the required size, performance, efficiency, and thermal characteristics of the main components in an electric powertrain.
Kulik et al. [7] estimate the requirements for a hybrid electric powertrain based on the analysis of a city vehicle GPS track together with accelerometer data. From the acceleration and velocity data of the track, the instantaneous power developed by the car is estimated and areas where regeneration is possible during braking.
Gearboxes are used for reasons of efficiency [8], in addition to improving the vehicle’s range and performance [9]. Different geared systems over the NEDC (New European Driving Cycle) cycle were tested and the energy consumption was calculated.
Transmissions with a gear ratio (i.e. a gearbox) have been optimized by Dagci et al. [10], among others. Standard drive cycles are used.
Guo et al. [11] propose a method for controlling the gearshift command in multispeed EV, reaching a reduction of 3% to 5% in energy consumption for city driving cycles. They used an algorithm, combining Pontryagin’s minimum principle and a numerical method, bisection method, for determining the gear positions and gearshift points. The algorithm was tested using the New European drive cycle (NEDC) and Urban Dynamometer Driving Schedule (UDDS).
Yu et al. [12], [13] minimize the time consumed by a vehicle to travel along a given route or track, both for 2 independent wheel driving and 4 independent wheel driving. Describing a very detailed vehicle model, they solve a large-scale nonlinear optimization problem constrained to a certain track. The base speed, the constant power speed ratio, the static braking force distribution and the gear ratio were selected as the design parameters of the vehicle. The methodology does not optimize the vehicle mass and do not consider the energy consumption.
Xin [14] develops an optimal design of the powertrain system in order to minimize the curb mass of the vehicle, due to the sensitivity of vehicle mass for energy consumption. To carry out the simulation the Urban Driving Cycle (UDC), the NEDC cycle and a constant speed cycle were chosen.
However, powertrain optimization is not the only way to improve the range of a vehicle. In order to improve the driving range of electric vehicles, especially for fleets of electric commercial vehicles, many studies [15] focus their efforts on planning the most efficient route. These studies use an average driver and are programmed to follow a driving cycle that include speed variation, topography, time stopped and changes in payload. They are focused in the track not in the vehicle or the travel time.
Other authors [16] focus their research on defining the best acceleration profile so that the energy consumed is minimized. Both a constant value of acceleration and a multi-step acceleration approach are considered in this last reference.
- Please mention the full form of any acronym used for the first time, e.g. NSGA2 or SPEA2.
The new version of the manuscript includes the following full forms:
New European drive cycle (NEDC)
Urban Dynamometer Driving Schedule (UDDS).
Urban Driving Cycle (UDC)
Multi-Objective Evolutionary Algorithms (MOEAs)
Non dominated Sorting Genetic Algorithm – 2 (NSGA2)
Strength Pareto Evolutionary Algorithm-2 (SPEA2)
Internal Combustion Engine (ICE)
Root Mean Square Error (RMSE)
Light Detection and Ranging (LIDAR)
- Please summarise the current limitations within the literature and need for the present work to address those limitations.
The new version of the manuscript includes the following sentences:
As can be seen, many research methods have been applied to optimize the powertrain of a specific vehicle and were validated according to a standard driving test cycle [26]. Therefore, the need to develop an optimization method that defines the optimal chassis design for an electric vehicle, not only for the powertrain system, but also for a specific route is evident.
The methodology proposed in this paper is intended not only to optimize the powertrain, but also to minimize energy consumption by determining the optimal chassis design for a given track, setting the travel time as an objective. This enables adapting the chassis design to the path selected by the driver, being applicable to transport vehicles that cover one or more fixed routes. On the other hand, a standard driving cycle is not used. A driving mode is defined that fully approves the performance of the vehicle based on the set path. This allows optimization of acceleration and braking cycles and speeds to minimize energy consumption.
In addition to other resistance forces normally considered, the vehicle model takes into account the effects of curves in order to improve the accuracy of the evaluation of energy consumption [27].
A previously set path and time allows the designer to find a solution adapted to the specific needs of the vehicle. The use of multi-objective genetic algorithms to solve this issue will serve as a reliable generic method for finding the optimal configuration of the chassis on any given route.
As a particular case, it will allow the design of gearboxes with an optimal relation for each route or the definition of an automatic change strategy depending on the chosen route.
- In the case study, the battery pack weight and the transmission ratio were optimised as vehicle chassis parameters. Please elaborate on the results obtained using your methodology with an existing method to demonstrate the usefulness of this method.
The new version of the manuscript includes the following sentences:
A design methodology based on standard driving cycles, without taking full advantage of the powertrain's performance, obtains mass values of the battery pack greater than 50 kg. If the methodology does not take into account the precise values of the route, but instead consider average values, the mass values of the battery pack would be greater than 42 kg. The implementation of methods of optimization of the consumed energy, without taking into account the multivariable analysis, does not allow obtaining values lower than 40 kg of the mass of the battery pack.
- Please elaborate the section 3 as Results and discussion. And then add a Conclusion section with future work.
The new version of the manuscript includes the following sentences:
This methodology has been used and validated in the optimization of a race car. The chosen track was a scoring event for the European Hill Climb Championship. In this case, the design variables were the battery pack mass and the transmission ratio, both optimized to ensure optimal performance on the track. The multi-objective model was validated with the use of a real-life case that confirms its functionality and accuracy. The implementation of the proposed methodology allows to reduce the weight of the battery pack by up to 20%, compared to traditional design methods.
For a more robust design, in future work the mathematical model should be able to cope with uncertainties in the acquired data and the state of the vehicle. We intend to deploy algorithms for multi-criteria optimization under uncertainty and obtain solutions are that are insensitive to small variations in weight of the loaded vehicle or the capacity of the batteries, to name some examples.

Reviewer 2 Report
This paper investigates the optimal design of the chassis of an electric vehicle. The main result is to present a multi-objective genetic algorithm determining the optimal chassis variable parameters considering the tradeoff among the mass of the vehicle, its energy consumption and the travel time. The topic is somewhat interesting. However, there are the following problems needing to be clarified or modified.
- The main contribution of the paper is not clear. Many research results have been presented on the optimal design of the powertrain for electric vehicles/hybrid electric vehicles using evolutionary algorithms, thus the contribution should be highlighted in Introduction.
- In Section 2.1 of optimization method, the flowchart shown in Fig.1 is simple and nothing new. the feature and advantage of the presented multi-objective genetic algorithm should be further explained and analyzed.
- In Section 3 of results, simulation verification is briefly given and is not enough to illustrate the feature and advantage of the proposed optimal method. The detailed analysis and illustration should be further given, including the comparisons with the existing closely related optimal methods.
- The paper is not well organized and the readability needs to be improved.
Author Response
Thank you in advance for your careful review of this paper. In this letter, we have reproduced your comments and have followed each comment with our response.
- The main contribution of the paper is not clear. Many research results have been presented on the optimal design of the powertrain for electric vehicles/hybrid electric vehicles using evolutionary algorithms, thus the contribution should be highlighted in Introduction.
The new version of the manuscript includes the following sentences in Abstract:
Abstract: An iterative algorithm is proposed for determining the optimal chassis design of an electric vehicle, given a path and a reference time. The proposed algorithm balances the capacity of the battery pack and the dynamic properties of the chassis, seeking to optimize the tradeoff between the mass of the vehicle, its energy consumption and the travel time. The design variables of the chassis include geometrical and inertial values, as well as the characteristics of the powertrain. The optimization is constrained by the slopes, curves, grip and posted speeds of the different sections of the track. Particular service constraints are also considered, such as limit accelerations due to passenger comfort or cargo safety. This methodology is applicable to any vehicle whose route and travel time are known in advance, such as delivery vehicles, buses and race cars, and has been validated using telemetry data from an internal combustion rear-wheel drive race car designed for Hill Climb competitions. The implementation of the proposed methodology allows to reduce the weight of the battery pack by up to 20%, compared to traditional design methods.
The new version of the manuscript includes the following sentences in Introduction:
Different design variables have been analyzed in relation to the different purposes of the optimization of the powertrain. Lei et al. [5] consider power performance, energy consumption and ride comfort using an in-wheel motor vehicle model as the basis for their multi-objective problem. They used two approaches to carry out the optimization, first a weighted objective method to transform the multi-objective optimization into a single-objective optimization and a second approach to obtain the optimal solution from the Pareto front. In this paper, energy consumption was evaluated by the urban driving cycle (ECE-15).
Domingues-Olavarría et al. [6] also take into account the cost of the powertrain as an additional constraint. To do it they used a model, simulated over a given drive cycle, that takes into account the required size, performance, efficiency, and thermal characteristics of the main components in an electric powertrain.
Kulik et al. [7] estimate the requirements for a hybrid electric powertrain based on the analysis of a city vehicle GPS track together with accelerometer data. From the acceleration and velocity data of the track, the instantaneous power developed by the car is estimated and areas where regeneration is possible during braking.
Gearboxes are used for reasons of efficiency [8], in addition to improving the vehicle’s range and performance [9]. Different geared systems over the NEDC (New European Driving Cycle) cycle were tested and the energy consumption was calculated.
Transmissions with a gear ratio (i.e. a gearbox) have been optimized by Dagci et al. [10], among others. Standard drive cycles are used.
Guo et al. [11] propose a method for controlling the gearshift command in multispeed EV, reaching a reduction of 3% to 5% in energy consumption for city driving cycles. They used an algorithm, combining Pontryagin’s minimum principle and a numerical method, bisection method, for determining the gear positions and gearshift points. The algorithm was tested using the New European drive cycle (NEDC) and Urban Dynamometer Driving Schedule (UDDS).
Yu et al. [12], [13] minimize the time consumed by a vehicle to travel along a given route or track, both for 2 independent wheel driving and 4 independent wheel driving. Describing a very detailed vehicle model, they solve a large-scale nonlinear optimization problem constrained to a certain track. The base speed, the constant power speed ratio, the static braking force distribution and the gear ratio were selected as the design parameters of the vehicle. The methodology does not optimize the vehicle mass and do not consider the energy consumption.
Xin [14] develops an optimal design of the powertrain system in order to minimize the curb mass of the vehicle, due to the sensitivity of vehicle mass for energy consumption. To carry out the simulation the Urban Driving Cycle (UDC), the NEDC cycle and a constant speed cycle were chosen.
However, powertrain optimization is not the only way to improve the range of a vehicle. In order to improve the driving range of electric vehicles, especially for fleets of electric commercial vehicles, many studies [15] focus their efforts on planning the most efficient route. These studies use an average driver and are programmed to follow a driving cycle that include speed variation, topography, time stopped and changes in payload. They are focused in the track not in the vehicle or the travel time.
Other authors [16] focus their research on defining the best acceleration profile so that the energy consumed is minimized. Both a constant value of acceleration and a multi-step acceleration approach are considered in this last reference.
[…]
The methodology proposed in this paper is intended not only to optimize the powertrain, but also to minimize energy consumption by determining the optimal chassis design for a given track, setting the travel time as an objective. This enables adapting the chassis design to the path selected by the driver, being applicable to transport vehicles that cover one or more fixed routes. On the other hand, a standard driving cycle is not used. A driving mode is defined that fully approves the performance of the vehicle based on the set path. This allows optimization of acceleration and braking cycles and speeds to minimize energy consumption.
In addition to other resistance forces normally considered, the vehicle model takes into account the effects of curves in order to improve the accuracy of the evaluation of energy consumption [27].
A previously set path and time allows the designer to find a solution adapted to the specific needs of the vehicle. The use of multi-objective genetic algorithms to solve this issue will serve as a reliable generic method for finding the optimal configuration of the chassis on any given route.
The new version of the manuscript includes the following sentences in Discussion:
Most published methods optimize the powertrain of electric vehicles in standard driving tests. In this contribution, a general methodology has been proposed that allows fine-tuning the chassis of electrical vehicles in an arbitrary route or set of routes. An approximate mathematical model has been developed that takes into account many different parameters of both the vehicle and track, such as geometries, inertial values, forces, resistances, grip and powertrain, among others. This set of information enables accurate evaluation of the consumed energy and at the same time is simple enough for being computationally efficient, thus it can be embedded in the fitness function of a multi-objective genetic algorithm.
This methodology has been used and validated in the optimization of a race car. The chosen track was a scoring event for the European Hill Climb Championship. In this case, the design variables were the battery pack mass and the transmission ratio, both optimized to ensure optimal performance on the track. The multi-objective model was validated with the use of a real-life case that confirms its functionality and accuracy. The implementation of the proposed methodology allows to reduce the weight of the battery pack by up to 20%, compared to traditional design methods.
- In Section 2.1 of optimization method, the flowchart shown in Fig.1 is simple and nothing new. the feature and advantage of the presented multi-objective genetic algorithm should be further explained and analyzed.
The new version of the manuscript includes the following figure caption:
Figure 1. General flowchart of the proposed methodology
- In Section 3 of results, simulation verification is briefly given and is not enough to illustrate the feature and advantage of the proposed optimal method. The detailed analysis and illustration should be further given, including the comparisons with the existing closely related optimal methods.
The new version of the manuscript includes the following sentences in Results:
A design methodology based on standard driving cycles, without taking full advantage of the powertrain's performance, obtains mass values of the battery pack greater than 50 kg. If the methodology does not take into account the precise values of the route, but instead consider average values, the mass values of the battery pack would be greater than 42 kg. The implementation of methods of optimization of the consumed energy, without taking into account the multivariable analysis, does not allow obtaining values lower than 40 kg of the mass of the battery pack.
- The paper is not well organized and the readability needs to be improved.
The new version of the manuscript was revised and improved (see submitted version).

Reviewer 3 Report
This manuscript proposed an iterative algorithm for determining the optimal chassis design of an electric vehicle, given a path and a reference time. The proposed algorithm balances the capacity of the battery pack and the dynamic properties of the chassis, seeking to optimize the tradeoff between the mass of the vehicle, its energy consumption and the travel time. The topic is important for the powertrain optimization of electric vehicles in terms of chassis design for a given track. However, there are some important points need further clarifying. Note that there are some obvious mistakes in this manuscript, it needs more careful editing.
Major comments:
- Formula derivation should be explained more clearly.
- The definition of parameters of engine and other power components should be more explicit, and it is better to have diagrams.
- The rationality of the approximate equation of formula 29 needs further explanation.
Minor comments:
- Starting from page 4, the number of lines is incomplete, so starting from page 4, the position of mistakes is indicated by the actual number of lines on this page.
- On the page 2, line 6 and line 8, the number of the formula of the braking acceleration should be (41).
- On the page 2, line 6 and line 8, “Equation (37)” should be “Equation (41)”.
- The first line is not indented in many places, such as the line 23 on the Page 5.
- On the page 9, line 19, there is an extra “:” behind the word “and”.
Author Response
Thank you in advance for your careful review of this paper. In this letter, we have reproduced your comments and have followed each comment with our response.
Major comments:
- Formula derivation should be explained more clearly.
The new version of the manuscript was revised and includes several cites, as. D. Gillespie, Fundamentals of Vehicle Dynamics, 1992 that explains formula derivation of Vehicle Dynamics and driving maneuvers
- The definition of parameters of engine and other power components should be more explicit, and it is better to have diagrams.
The new version of the manuscript was revised and includes a figure 3 as follows:
Figure 3. Torque and power curves of an electric motor.
- The rationality of the approximate equation of formula 29 needs further explanation.
The rationality of equation of formula 29 can be seen in:
Guo, L.; Gao, B.; Liu, Q.; Tang, J.; Chen, H. On-line optimal control of the gearshift command for multispeed electric vehicles. IEEE/ASME Transaction on Mechatronics 2017, vol. 22 (4), pp. 1519–1530. DOI: 348 10.1109/TMECH.2017.2716340.
Minor comments:
- Starting from page 4, the number of lines is incomplete, so starting from page 4, the position of mistakes is indicated by the actual number of lines on this page.
The new version of the manuscript was revised and theses problems were solved.
- On the page 2, line 6 and line 8, the number of the formula of the braking acceleration should be (41).
The new version of the manuscript was revised and includes that sentences:
In some cases, however, the braking acceleration (41) might not be suitable for passenger comfort or cargo safety. Hence, depending on the case, a maximum longitudinal acceleration (a_x) should be established that is compared with the value obtained in Equation (41). The minimum value is consequently chosen;
- The first line is not indented in many places, such as the line 23 on the Page 5.
The new version of the manuscript was revised and modified.
- On the page 9, line 19, there is an extra “:” behind the word “and”.

Round 2
Reviewer 2 Report
Comparison with the original version, the revision of the paper submitted is suitable for publication. No comments.
Author Response
Thank you in advance for your careful review of this paper.
Reviewer 3 Report
This manuscript proposed an iterative algorithm for determining the optimal chassis design of an electric vehicle, given a path and a reference time. The contributors responded to the major comments and revised the raised minor comments in the manuscript. The revised manuscript is more explicit in content and corrects many editorial errors. However, there are still some points in this manuscript that need to be further clarified. And some minor editing errors still need to be corrected.
Major comments:
- Explain the function of Figure 4 or the useful information obtained from Figure 4.
Minor comments:
- On the page 10, line 283, there is still an extra “:” behind the word “and”
Author Response
Thank you in advance for your careful review of this paper. In this letter, we have reproduced your comments and have followed each comment with our response.
Major comments:
- Explain the function of Figure 4 or the useful information obtained from Figure 4.
The new version of the manuscript includes the following sentences:
In the proposed methodology, the Geographic Information System (GIS) data is pre-processed before modeling. The road centerline (Figure 4) is determined by analyzing the GIS data and orthophotos. Subsequently, the precision of the route is improved and adapted to the terrain by combining the horizontal alignment and elevation profile.
The new version of the manuscript includes the following sentences in figure caption:
Figure 4. 3D characterization of the track. The road centerline (red) is shown in a Geographic Information System using a digital model (yellow) of the study area and road
Minor comments:
- On the page 10, line 283, there is still an extra “:” behind the word “and”
The new version of the manuscript was revised and solved that mistake (see submitted version).
